# Ultra Stable Molecular Sensors by Submicron Referencing and Why They Should Be Interrogated by Optical Diffraction—Part II. Experimental Demonstration

**DOI:** 10.3390/s21010009

**Published:** 2020-12-22

**Authors:** Andreas Frutiger, Karl Gatterdam, Yves Blickenstorfer, Andreas Michael Reichmuth, Christof Fattinger, János Vörös

**Affiliations:** 1Laboratory of Biosensors and Bioelectronics, Institute for Biomedical Engineering, University and ETH Zürich, 8092 Zürich, Switzerland; frutiger@biomed.ee.ethz.ch (A.F.); blickenstorfer@biomed.ee.ethz.ch (Y.B.); reichmuth@biomed.ee.ethz.ch (A.M.R.); 2Institute of Structural Biology, University Hospital Bonn, University of Bonn, 53127 Bonn, Germany; karlgatterdam@uni-bonn.de; 3Roche Pharma Research and Early Development, Roche Innovation Center Basel, 4070 Basel, Switzerland

**Keywords:** label-free biosensors, optical diffraction, shot noise limit, focal molography

## Abstract

Label-free optical biosensors are an invaluable tool for molecular interaction analysis. Over the past 30 years, refractometric biosensors and, in particular, surface plasmon resonance have matured to the *de facto* standard of this field despite a significant cross reactivity to environmental and experimental noise sources. In this paper, we demonstrate that sensors that apply the spatial affinity lock-in principle (part I) and perform readout by diffraction overcome the drawbacks of established refractometric biosensors. We show this with a direct comparison of the cover refractive index jump sensitivity as well as the surface mass resolution of an unstabilized diffractometric biosensor with a state-of-the-art Biacore 8k. A combined refractometric diffractometric biosensor demonstrates that a refractometric sensor requires a much higher measurement precision than the diffractometric to achieve the same resolution. In a conceptual and quantitative discussion, we elucidate the physical reasons behind and define the figure of merit of diffractometric biosensors. Because low-precision unstabilized diffractometric devices achieve the same resolution as bulky stabilized refractometric sensors, we believe that label-free optical sensors might soon move beyond the drug discovery lab as miniaturized, mass-produced environmental/medical sensors. In fact, combined with the right surface chemistry and recognition element, they might even bring the senses of smell/taste to our smart devices.

## 1. Introduction

Label-free interaction analysis is important in drug discovery and basic research in molecular biology [1,2]. The prominent classes of sensors in these fields are temperature stabilized and reference subtracted refractometric (bio)sensors, mainly based on surface plasmon resonance [3]. Surface plasmon resonance (SPR) is widespread because of a favorable combination of sensitivity, a relative simple readout method and chip fabrication, and historical reasons [4]. Amongst the most sensitive refractometric configurations are also interferometric refractometric biosensors [5,6,7,8]. A few interferometric biosensors based on planar waveguide technology such as grating coupled or dual-polarization interferometry exist or have existed as commercial devices [9,10,11,12,13]. However, integrated optical refractometric biosensors (e.g., Young and Mach–Zehnder interferometers, bimodal sensors, ring resonators and photonic crystal sensors) have so far not succeeded in commercialization. This is due to a more elaborate fabrication process and a more complex readout compared to SPR (tunable laser source, on-chip spectrometers or phase modulators) [14]. In addition, no silicon photonics devices that operate at telecommunication wavelengths (1310 and 1550 nm) are on the market because they are severely limited by water absorption [15]. Refractometric biosensors operating in the visible range achieve impressive short time resolutions of 10${}^{-8}$ refractive index units (RIU) (approximately 10 fg/mm${}^{2}$) [16]. In a more figurative sense, this corresponds to the detection of only ∼0.003% of a monolayer of water molecules on a dry sensor surface. However, refractometric biosensors are also extremely cross-sensitive to refractive index changes of the cover medium, to temperature drifts and to nonspecific binding to the sensor surface [1,17,18,19,20,21,22,23]. This is because refractometric biosensors solely integrate the refractive index over the entire sensing volume. Therefore, most refractometric biosensors can only be operated under stable conditions (continuous flow of buffer and temperature stabilization). This precludes any meaningful miniaturization of these devices [24]. However, even when stabilized, they experience significant drift, which over the timescale of an experiment limits the resolution to 1–5 μRIU (1–5 pg/mm${}^{2}$) [25]. On the other hand, diffractometric biosensors were demonstrated to be largely unaffected by these extrinsic influences, even in the complete absence of stabilization [26,27]. The reason for this stability lies in the intrinsic architecture of the diffractometric biosensors and in the arrangement of the binding sites in particular. By design, a diffractometric biosensor constrains the analyte molecules to bind in the form of submicron periodic grating. In other words, the analyte-induced refractive index increase is modulated in space. This essentially creates interdigitated signal and reference regions on the submicron scale, which allow for efficient self-referencing. In light of the spatial lock-in concept described in part I, the modulation shifts the signal power (amount of analyte bound) to the spatial frequency of the grating (inverse of the grating period) and efficiently separates it from the environmental noise [28]. Environmental noise such as temperature gradients are long-ranged and therefore situated at low spatial frequencies [28]. In contrast to an integrative refractometric biosensor, the phenomenon of diffraction creates a sensor transfer function that only measures the Fourier components of the refractive index distribution with a spatial period close or equal to the grating period [28]. More precisely, diffractometric biosensors constitute a high-frequency spatial affinity lock-in. A lock-in can detect signals buried in a noisy background that is orders of magnitude larger [29]. We thoroughly introduced the concept of the spatial affinity lock-in and discussed its similarities with the time domain lock-in in part I [28,29].

In this paper, we experimentally demonstrate the intrinsic robustness of a spatially locked-in sensor. In addition, we provide additional arguments to the ones already presented in part I on why diffraction has fundamental advantages over refraction for the readout of a locked-in binding signal. For this purpose, we developed a combined waveguide-based diffractometric/refractometric biosensor. Combined refractometric diffractometric biosensors are not a new concept, and they have already been demonstrated with surface plasmon diffraction [30]. However, they were not yet thoroughly investigated in a quantitative manner, mainly because of a lack of understanding of the quantification of diffractometric biosensors until a few years ago. We recently derived a theory of quantitative diffractometric biosensors [31]. We are now in a position to perform quantitative comparison of a combined refractometric and diffractometric biosensor that is subject to the same experimental conditions. At this example and in combination with measurements on a stabilized and reference subtracted commercial biosensor (Biacore 8k), we show that a locked-in sensor is considerably more resilient to the environment and disturbing experimental influences. Only a high-frequency spatially locked-in sensor can efficiently reject refractive index discontinuities as they occur, for instance, by buffer changes [32]. Indeed, we demonstrate that the correlation between the interdigitated sensing and reference regions upon a refractive index jump is so perfect that the remaining sensor response is simply caused by the refractive index change in the displacement volume of the bound molecules themselves. The resilience of the locked-in sensor to bulk refractive index changes enables investigations in label-free biosensing that are difficult or not possible with refractometric methods like SPR and waveguide interferometry [26,33,34]. Furthermore, we will see that, also in terms of resolution and long-term drift stability, a non-stabilized locked-in sensor is superior to a stabilized and reference subtracted commercial device.

In the second part, we experimentally and theoretically show that, in order to achieve the same mass resolution, a refractometric biosensor requires relative measurement precision that is up to four orders of magnitude higher than that of a diffractometric biosensor. One entire section is devoted to the conceptual explanation of the underlying physical reason for this effect. In essence, it stems from two fundamental differences in refraction and diffraction: First, in refractometric biosensors, the binding of molecules is connected to the phase of the applied electromagnetic field. Conversely, in diffractometric biosensors, it is related to the square amplitude of the diffracted field (i.e., its intensity). Phase measurements require a phase reference. Therefore, phases can only be measured by an indirect detection scheme (homodyne or heterodyne detection) or via a resonance condition (guided mode resonance [35] and localized surface plasmon resonance [36]). This is because detectors at optical frequencies, i.e., a photodiode, can only detect the magnitude of a phasor. Therefore, direct detection schemes, i.e., without a reference phasor, can only be applied if binding information is connected to the magnitude of the phasor. Direct and indirect detection are fundamentally different when it comes to noise and stabilization requirements. Indirect detection implies high measurement precision and sophisticated stabilization of the phase reference. Therefore, the figure of merit of refractometric sensors is limited to around 100/RIU. Given that, it is difficult to find the center position of a broad resonance in resonance-based refractometric sensors or to stabilize the reference phasor in an interferometric refractometric sensor. Conversely, the limiting factor of a diffractometric biosensor is not the measurement precision but rather the magnitude of the stray light phasor. The background is inherently low because the off-axis detection eliminates any interference effects stemming from the zeroth diffraction order. Furthermore, optical surfaces can be extremely smooth and the remaining irregularities are random. Therefore, the stray light phasor only corresponds to a coherent mass density of about 1 pg/mm${}^{2}$. We will learn that it is mainly this “short” stray light phasor that leads to a typical figure of merit of diffractometric biosensors of 500,000/RIU. Such values have never been matched even closely with ever so sophisticated refractometric biosensors. The second fundamental advantage is that most refractometric biosensors (surface plasmon resonance and interferometric biosensors) accumulate the phase shift induced by the higher refractive index of the bound molecules over a certain propagation distance. Diffractometric biosensors detect unbalanced refractive index changes over one wavelength of the oscillating electromagnetic field. In other words, diffractometric biosensors are an almost perfectly balanced local common path interferometer that is extremely stable against mechanical vibrations and long-range refractive index fluctuations [37]. The advantages of a local common path interferometer, a small diffracted phasor, Fourier domain and off-angle detection for improvements in signal to noise have already been described by Wang et al. [37]. However, their “diffractive optical balance” has one significant drawback. Although based on detection of an off-angle diffraction order, the device was a pure refractometric sensor without any spatial lock-in. Inorganic material on the ridges of the diffractive optical balance displaces some of the volume water molecules from the volume illuminated by the evanescent field. Therefore, its sensing response is essentially the cross-sensitivity of a diffractometric sensor to cover index changes. Luckily for diffractometric sensors, this refractometric response is very small compared to the actual diffractometric response, as we shall see in this paper.

In summary, this paper shows that the spatial lock-in principle solves the cross-sensitivity problem of traditional label-free optical biosensors that are based on the refractometric principle. The consequence of this insight is significant: Label-free optical biosensors can be robust, sensitive, small and simple at the same time. This might enable completely new application areas of label-free sensing.

## 2. Results and Discussion

### 2.1. A Combined Refractometric and Diffractometric Biosensor

The combined refractometric/diffractometric biosensor that will be used for a direct comparison between the sensing principles is essentially an outcoupling grating on a single-mode waveguide.

Refractometric biosensors based on outcoupling gratings have been described 35 years ago by the group of Lukosz [38,39,40,41,42]. Such a sensor can detect binding events because the relatively high refractive index of biomolecules compared to water locally increases the refractive index of the cover medium. This in turn affects the effective refractive index of the guided mode (change in the phase velocity) and finally leads to an increase in the outcoupling angle of the grating via the grating equation [43]:(1)$$N=n_{c}\sin\left(\theta \right)+\frac{m\lambda }{\Lambda },$$
where *N* is the effective refractive index of the guided mode, $n_{c}$ is the refractive index of the cover medium, θ is the outcoupling angle with respect to the surface normal, *m* is the diffraction order, λ is the wavelength and Λ is the grating period.

Sensitive diffractometric biosensors that are based on outcoupling gratings are relatively new and offer exciting new applications [26,27]. In particular, they enable real-time label-free measurements in complex fluids as well as the label-free observation of signaling cascades within living cells [33,34]. These sensors detect binding events by a change in the diffraction efficiency of the outcoupling grating due to binding of the molecules in a coherent, i.e., ordered, fashion. Compared to previous implementations of diffractometric biosensors [44,45], the arrangement described in [26,27] has substantially improved stability and performance. This is due to multiple reasons [46]. First, the spatial modulation frequency is considerably higher than previously because the grating period is submicron [26,27,46]. This achieves an effective separation of signal and environmental noise by a high-frequency spatial affinity lock-in [28]. Second, the grating lines composed of the binding sites are not straight and equidistant but rather chirped and curved to form a focusing grating coupler (diffractive lens) [26,27,46,47,48,49,50]. The lens constructs the Fourier plane necessary for the analog spatial lock-in in close proximity to the chip without any additional optical components [27,28,46]. Third, contrary to previous transmission or reflection-based diffractometric approaches, the evanescent wave of the guided mode only illuminates the first 100 nm of the sample, i.e., it provides darkfield illumination [26,27,46,51,52]. Before the introduction of focal molography [46], the performance of diffractometric sensors was severely limited because of suboptimal design in all of these three parameters. The class of diffractometric biosensors that incorporates a high spatial frequency affinity lock-in, a diffractive lens and darkfield illumination was termed focal molography [46]. The name originates from a combination of the words molecule and holography given that the analyte molecules form a focusing hologram in which the Airy disk intensity serves as the sensor output [26,27,33,46,50].

The basic sensor arrangement was described in detail in a previous publication [27] as well as is briefly summarized in Figure 1f and Appendix G. The blueprint of the molecular hologram has the shape of a focusing outcoupling grating which consists of roughly 1000 slightly curved and chirped grating periods. Each period has an analye binding region (“ridge”) and a reference region (“groove”) (Figure 1a). It is illuminated by the fundamental TE (transverse electric) mode of a high refractive index single-mode slab waveguide. For the combined refractometric and diffractometric measurement, the grating is slightly biased by covalently bound polystyrene particles. This creates a visible focal spot already at the start of the measurement (Figure 1b). Molecules binding to the ridges scatter additional electromagnetic intensity which constructively interferes in the focal spot. This leads to an increased intensity in the Airy disk of the mologram (Figure 1c). This is the diffractometric response of the sensor. From the diffraction efficiency $\frac{P_{\mathrm{out}}}{P_{\mathrm{in}}}$, one can quantify the coherent mass density on the sensor:(2)$$\Gamma_{\mathrm{coh}}=\frac{1}{2\left|K\right|n_{c}\frac{dn}{dc}A}\sqrt{\frac{P_{\mathrm{out}}}{P_{\mathrm{in}}}},$$
where *K* is the coupling coefficient [31], *A* is the area, $\frac{dn}{dc}$ is the refractive index increment for proteins in water [46], $n_{c}$ is the refractive index of the cover medium and $\Gamma_{\mathrm{coh}}$ is the coherent mass density. The total mass density $\Gamma_{\mathrm{tot}}$ can be obtained from $\Gamma_{\mathrm{coh}}$ by division with the analyte efficiency $\eta_{\left[A\right]}$, which is discussed in [27,31,50]. The coupling coefficient for an incident-guided TE mode and outgoing free space mode into the substrate reads $\left|K\right|=\left|\frac{\sqrt{2}\pi }{\lambda \sqrt{n_{s}N}}\sqrt{\frac{\left(n_{f}^{2}-N^{2}\right)}{\left(n_{f}^{2}-n_{c}^{2}\right)}}\frac{1}{\sqrt{A}\sqrt{w_{b}t_{\mathrm{eff}}}}\frac{t_{s}^{s}\left(\theta_{\mathrm{out}}\right)}{\cos\left(\theta_{\mathrm{out}}\right)}\right|$, with $w_{b}$ being the beam width of the guided beam, $t_{\mathrm{eff}}$ being the effective thickness of the guided mode, *N* being the effective refractive index, $n_{f}$ being the refractive index of the waveguiding film, $t_{s}^{s}\left(\theta_{\mathrm{out}}\right)$ being the angle-dependent Fresnel transmission coefficient of the three layer interface [53] and $\theta_{\mathrm{out}}$ being the angle with respect to the surface normal of the outgoing beam [31]. In this paper, we assumed a perpendicular outgoing beam ($\theta_{\mathrm{out}}=0$), which is a good approximation since the numerical aperture of the chirped outcoupling grating is small.

Any molecule that binds to the surface also shifts the focal spot in lateral position (Figure 1d). The movement is in the mode propagation direction when the mass density increases and opposite when it decreases. This is the refractometric response of the sensor and the focal spot shifts because the phase shift leads to change in outcoupling angle, as described above with Equation (Equation 1). If the binding happens to be uniform (not coherent), the intensity of the focal spot hardly changes and the response is purely refractometric. The intensity does not change because the molecules on the grooves interfere destructively and cancel the contribution from the ridges. In order to obtain the change in surface mass density ΔΓ from the shift of the molographic focus *x*, one needs to derive the sensitivity of the surface mass density change with respect to the shift in the focal plane $\frac{\partial \Gamma }{\partial x}$ (Appendix B).
(3)$$\Delta \Gamma =\frac{1}{\frac{dn}{dc}f}\frac{t_{\mathrm{eff}}}{2}\frac{N}{n_{c}}\frac{n_{f}^{2}-n_{c}^{2}}{n_{f}^{2}-N^{2}}\Delta x$$
where *f* is the focal distance of the mologram in air. Appendix A shows how the focal spot shifts when the cover medium changes from air to water.

We implemented the two readout methods, a diffractometric and a refractometric mass readout, in Python, and the exact algorithm is described in the methods section as well as in [27]. Figure 1c,e shows the diffractometric and refractometric responses of the sensor when 100 nM SAv (Streptavidin) is injected as a proof of principle experiment after 700 s of baseline recording in PBS-T (phosphate buffered saline with Tween20). The molograms for this experiment were functionalized with biotin on the ridges and polyethylene glycol on the grooves ([NH-biotin|NH-PEG${}_{12}$]) (see the Materials and Methods section). As mentioned in the introduction, there is an important difference between the two sensor outputs: the diffractometric output is inherently self-referencing (locked-in) with a very high spatial frequency, whereas the refractometric output integrates the change in refractive index over the entire evanescent field volume. Thus, the refractometric biosensor will pick up much more environmental refractive index noise. This is already visible if one compares the noise of Figure 1c,e.

### 2.2. Experimental Noise Rejection Performance of a Spatial Lock-In—Refractive Index Jumps of the Cover Medium

As an example of *experimental noise rejection*, we demonstrate the insensitivity of the spatial lock-in principle to refractive index changes in the sensing volume. Insensitivity of a biosensor to refractive index changes is of considerable interest for drug screening because most drug candidates are stored as DMSO (dimethylsulfoxide) solutions (refractive index *n*≈ 1.5). This constitutes a real challenge for refractometric sensors, especially in assays where a reference channel cannot be used [54]. For this reason, we also investigate the refractive index change susceptibility of unreferenced refractometric biosensors.

#### 2.2.1. Sensitivity of Unreferenced Refractometric Biosensors to Cover Index Changes

To get an estimate of the effect of cover index changes on the sensor output, one needs to derive the sensitivity of the sensor output to refractive index changes in the cover medium. A typical sensor output of a refractometric sensor is the surface mass density. For the case of an unreferenced refractometric biosensor, the sensitivity of the mass quantification to cover index changes has already been derived by Lukosz [55] and reads
(4)$$\frac{\partial \Gamma }{\partial n_{c}}=\frac{\Delta z_{c}}{2\frac{dn}{dc}},$$
with $\Delta z_{c}$ being the penetration depth of the evanescent field. Such a sensor outputs a change in surface mass density for a refractive index change of the cover of $\Delta n_{c}$ according to $\Delta \Gamma =\frac{\Delta z_{c}}{2\frac{dn}{dc}}\Delta n_{c}$. For our refractometric sensor, the sensitivity is 225,000 pg/mm${}^{2}$/RIU. To demonstrate this experimentally, we applied six refractive index standard solutions (glycerol/water mixture) to a sensor that was slightly biased by polystyrene particles on the ridges (see the Materials and Methods section). Already refractive index jumps in the order of 0.01 RIU led to sensor output changes typical for a monolayer of protein molecules (2250 pg/mm${}^{2}$) (Figure 2a). [56] Figure 2c shows the measured output change vs. the one predicted from the sensitivity to refractive index changes. The sensitivity to refractive index changes scale with penetration depth. A sensor based on surface plasmon resonance (320 nm penetration depth [3]) is roughly four times more susceptible to refractive index jumps than the waveguide-based sensor discussed here (82 nm pentration depth). This can be seen nicely by comparing the response in pg/mm${}^{2}$ in Figure 2a and Figure 3a for the same refractive index discontinuities. (As a side remark, a multitude of approaches have been developed to discriminate between the bulk refractive index changes and surface effects, for instance, by a penetration depth difference of two surface plasmon modes [57,58,59]. In essence, this is just a clever way to artificially reduce the penetration depth by postprocessing. Unfortunately, refractive index changes that happen at the surface (due to nonspecific binding) cannot be referenced out by this approach.) In summary, an unreferenced refractometric biosensor experiences output changes in the order of a monolayer of protein molecules for the applied refractive index jumps.

#### 2.2.2. Sensitivity of Diffractometric Biosensors to Cover Index Changes

In contrast, the diffractometric output is hardly affected (Figure 2b). One can see that, even for a 0.03 RIU refractive index discontinuity, the signal only decreases by 2–3 pg/mm${}^{2}$. This is more than three orders of magnitude less than the refractometric response in the very same experiment. Equation (Equation 5) describes the sensitivity of the cover index change for a diffractometric sensor (see Appendix C for the derivation)
(5)$$\frac{\partial \Gamma_{\mathrm{coh}}}{\partial n_{c}}=\Gamma_{\mathrm{coh}}\frac{1}{n_{c}}\left(1+\frac{\left(n_{P}^{4}-5n_{P}^{2}n_{c}^{2}-2n_{c}^{4}\right)}{n_{P}^{4}+n_{P}^{2}n_{c}^{2}-2n_{c}^{4}}\right).$$
This equation assumes that the refractive index jump is perfectly correlated over one signal and reference region. This is justified since the grating period is only in the order of 350 nm and since equilibration by advection over this length scale happens within a fraction of a second [26]. Diffractometric sensors have a small sensitivity to cover index changes because the molecules/particles on the signal region occupy only a tiny physical volume of the evanescent field. In the experiment, the sensor was slightly biased with polystyrene particles to a coherent mass density $\Gamma_{\mathrm{coh}}$ of 23.4 pg/mm${}^{2}$. If we insert this value and, additionally, $n_{P}=1.588$ for polystyrene particles, in Equation (Equation 5), we obtain −69.3 pg/mm${}^{2}$/RIU for the sensitivity to cover index changes in water. This is three orders of magnitude less than for the unreferenced refractometric sensor. We plotted this as a function of the applied refractive index discontinuity in Figure 2d together with the experimental values from Figure 2b. One can see that the model and the experimental data agree well. It can thus be concluded that high-frequency spatially locked-in biosensors can almost fully reject refractive index jumps. Furthermore, the readout is only affected because of the molecules already present on the sensor, i.e., the capture molecules. This is a property of any referenced sensor, as we shall see in the next paragraph.

#### 2.2.3. Sensitivity of Commercial Referenced Refractometric Sensors to Cover Index Changes

A diffractometric biosensor has reference and signal regions so close together that a refractive index jump is perfectly correlated and only the molecules on the sensor cause a response. An interesting question is how this compares to a reference subtracted and stabilized refractometric biosensor. In general, the referenced refractometric sensor output when no molecules are bound is $\Delta \Gamma \;=\;\frac{1}{2\frac{dn}{dc}}\left(\Delta z_{c,\mathrm{sig}}\Delta n_{c,\mathrm{sig}}-\Delta z_{c,\mathrm{ref}}\Delta n_{c,\mathrm{ref}}\right)$. Thus, not only the refractive index discontinuities in signal and reference regions need to be correlated but also the penetration depths must be the same at both locations. In commercial refractometric biosensors, the reference and signal channels are usually in a separate flow cell and a few hundred microns to even millimeters apart. It is obvious that good correlation between signal and reference drifts/jumps and in the penetration depths will be difficult to achieve for millimetre-scale separation distances. This is because thin films are generally not uniform. In addition, any gradient in refractive index equilibrates at a finite speed by the advection diffusion equation [43,60]. Boecker et al. nicely demonstrated that correlation progressively increases for decreasing separation distances (higher spatial lock-in frequency) between signal and reference regions and that the sensor becomes less prone to noise [32]. The distributed referenced refractometric sensors are an example of a digital lock-in amplifier (as introduced in part I) [28]. Also, here, it can be shown that, in the high spatial frequency limit, the sensitivity to refractive index jumps is solely due to the mass difference on the signal and reference regions (Appendix D)

To demonstrate that a commercial referenced and stabilized refractometric biosensor cannot reference refractive index jumps as efficiently as a spatially locked-in sensor, we conducted the same refractive index discontinuity experiment on a Biacore 8k instrument (GE Healthcare). The results are summarized in Figure 3. Figure 3a shows that the unreferenced sensor output changes by multiple protein monolayers. Even if referenced (Figure 3b), there are considerable artefacts (in the order of a few 1000 pg/mm${}^{2}$) at the beginning and the end of the injection due to time delays of the injected fluid. This is followed by a transient response of roughly 50 s (Figure 3c). After that time, the refractive index difference between sensing and signal channels is likely equalized. However, due to different penetration depths (small variation in gold thickness, etc.) the sensor output is not the same. Figure 3d displays the differential sensor output for three experiments in three different channels as a function of the applied refractive index discontinuity. One can estimate the sensitivity of the referenced sensor to refractive index discontinuities by linearly fitting a line (grey dashed line) through the data in Figure 3d, which yields 760 pg/mm${}^{2}$/RIU. This is nearly an order of magnitude larger than the cross-sensitivity of the spatially locked-in diffractometric sensor discussed before.

### 2.3. Environmental nOise Rejection Performance of a spatial lock-in—Resolution of Unstabilized Spatially Locked-In Biosensors Compared to Stabilized and Reference Subtracted Commercial Biosensors

Experimental noise rejection, such as insensitivity to refractive index discontinuities, is not the only necessary feature of a robust biosensor. In this section, we *investigate the environmental noise rejection* capabilities of spatially locked-in biosensors compared to stabilized and reference subtracted devices. The quantity that characterizes the environmental noise rejection capability is the mass density resolution of the sensor. The mass density resolution of a biosensor is defined as the standard deviation of the sensor output $\sigma_{\mathrm{so}}$ divided by the sensitivity of the sensor output $S_{\Gamma }^{\mathrm{so}}$ with respect to the mass density Γ (in the case of diffractometric sensors the coherent mass density $\Gamma_{\mathrm{coh}}$ [27,31]):(6)$$\sigma_{\Gamma }=\frac{\sigma_{\mathrm{so}}}{S_{\Gamma }^{\mathrm{so}}}.$$

In order to compare the two systems, we acquired baselines under a constant flow of buffer (see the Materials and Methods section). Figure 4a shows the referenced Biacore baselines compared to the locked-in baseline. The best and the worst performing channels out of the eight channels of the 8k+ are displayed for each experiment. It can be observed in Figure 4a that all of the worst performing refractometric channels experience a drift between 0.5–2 pg/mm${}^{2}$ over the time course of three hours. Some of the best performing channels are on par with the spatially locked-in sensor, which experiences virtually no drift over the entire measurement time (below 100 fg/mm${}^{2}$). However, the drift is not a particularly rigorous method to compare the noise performance of systems. For this reason, we computed the power spectral density of the binding traces. Figure 4b shows the power spectral density of the mass density noise as a function of frequency. Here, it is immediately apparent that the spatially locked-in sensor exhibits lower noise levels over the entire frequency spectrum than the worst performing referenced refractometric channels but, in particular, at low frequencies (long experiment durations). The power spectral density (PSD) also shows that the best referenced refractometric measurements are comparable to the locked-in sensor. Power spectral densities are rather difficult to read for people outside the signal processing domain. Therefore, we give a more intuitive description of the noise by integrating the PSD from the sampling frequency to the frequency that corresponds to the experiment duration. This yields the total noise picked up by the system over this time interval, which corresponds to the experiment duration. This total noise is then plotted in Figure 4c as a function of the data acquisition time. Also, in this form, it is obvious that only the best referenced refractometric sensing traces have similarly low noise levels as locked-in sensors. This might not seem too exciting at first, but one has to be aware that we compared a completely unstabilized and nonoptimized locked-in system with one of the best temperature stabilized commercial surface plasmon resonance systems with over more than 30 years of engineering experience. Even worse, only by incorporating an additional flow cell and a sophisticated temperature control into the system does the refractometric measurements become as robust as locked-in sensors. To demonstrate this, we have also included the individual responses of the two Biacore flow cells in the appendix (Figure A3). The noise of these baselines is roughly one order of magnitude higher than that of the referenced flow cell. This further exemplifies the superior robustness of the locked-in sensing concept compared to traditional refractometric sensors.

### 2.4. Baseline Noise of a Refractometric and a Diffractometric Biosensor under the Same Measurement Conditions

In the previous sections, we have shown that a locked-in sensor is more robust than a state-of-the-art, commercial, reference subtracted biosensor. In this section, we will investigate the resolution of a refractometric and a diffractometric biosensor under the same environmental and measurement conditions. This shall emphasize the impact that temperature stabilization and 30 years of engineering had on the resolution of refractometric biosensors and how forgiving the diffractometric sensing principle is in this regard. The experiment shall further serve as the motivation for the remainder of this paper. It is not only the spatial lock-in and Fourier space detection that make diffractometric biosensors more robust. It is intrinsic to their sensing principle that they have considerably lower requirements on the precision of the readout instrument than refractometric sensors to achieve the same baseline resolution.

For the comparison, we acquired baselines of the simultaneous refractometric and diffractometric sensor outputs (see the Materials and Methods section) in a buffer. Figure 5a,b shows the refractometric as well as the diffractometric baselines. The refractometric sensing baseline is roughly a factor 1000 noisier than the diffractometric one (100 pg/mm${}^{2}$ vs. 0.1 pg/mm${}^{2}$ rms noise). The noise source that compromises the refractometric resolution is primarily mechanical (Appendix H). In fact, the refractometric readout is extremely susceptible to mechanical drifts (thermal expansion etc.) (Equation (Equation 3)). To give an example, only a shift of 0.7 nm in the focal plane is required to change the sensor output by 1 pg/mm${}^{2}$ in the case of a 2-mm focal distance mologram. This can be ameliorated by increasing the focal distance (Figure A2). However, to achieve tolerances in the μm range for resolutions of 1 pg/mm${}^{2}$, one would have to measure tens of centimeters away. This would result in a bulky readout instrument. On the other hand, thanks to an array detector and image registration (see the Materials and Methods section), the diffractometric output is virtually insensitive to lateral and angular drifts. The longitudinal drifts are also negligible because the elongation of the Airy disk ($\Delta z=\frac{2n_{s}\lambda }{NA^{2}}$[53], where $n_{s}$ is the refractive index of the medium at the position of the Airy disk) spans several tens of micron already for numerical apertures (NA) of 0.1. Thus, diffractometric biosensors are much more misalignment-tolerant than any simple angle-based refractometric sensor. This is insofar important since most state-of-the-art commercial refractometric biosensors rely on an extremely precise angle measurement.

### 2.5. Fundamental Differences of Diffractometric and Refractometric Biosensors in Terms of Required Measurement Precision

#### 2.5.1. Conceptual Discussion

Mechanical drift sensitivity just represents one example of an experimental influence. In fact, as mentioned in the Introduction section, the difference in measurement error tolerance between refractometric and diffractometric biosensors is fundamental and generalizable. In part I [28], we already elucidated on some of these aspects in the discussion on analog (diffractometric sensor) and digital (distributed referenced refractometric) lock-in amplifiers. In particular, we emphasized the importance of detecting the sensor signal in Fourier space for reducing measurement noise. However, this is only part of the implication. In refractometric sensors, the binding event to the sensor surface is connected to the phase shift of the incident beam or the shift of a resonance, while in diffractometric sensors, binding creates a diffracted beam that is absent or very weak before binding. To understand this second fundamental advantage, it is helpful to think of optical biosensors as interferometers and to discuss their operation principle in the complex plane with the help of phasors. For explanation purposes, we will discuss interferometric refractometric biosensors in the next sections and devote one paragraph to resonance-based refractometric biosensors. When a beam of coherent light passes through a sample with diluted biomolecules, each of these molecules acts as a scatterer that produces a spherical wave which is much weaker than the incident beam (Figure 6a). In the forward direction, these scattered waves interfere with the incident beam in quadrature in the far field (Chapter 2.4 in [61]). Thus, the only effect of the molecules is phase retardation of the incident beam (a rotation of the incident phasor by the scattered phasors). This is known as refraction. On the other hand, any light that is scattered in off-axis directions or backwards is separated from the incident beam. Hence, in these directions, only stray light phasors interfere in the far field. Since their phases are uniformly distributed in the range (−π, π), they form a random phasor sum with a resultant that has a length that is Rayleigh distributed [62]. The first moment of this phasor (${\bar{A}}_{\mathrm{res}}=1.25a_{n}\sqrt{N_{a}/2}$, where $a_{n}$ is the length of one phasor and $N_{a}$ is the number of phasors) is only proportional to the square root of the number of molecules. If, however, the molecules are placed on a surface such that their phases interfere constructively in one angular direction, then the resultant’s length is proportional to the number of molecules (${\bar{A}}_{\mathrm{res}}=a_{n}N_{a}$) (Figure 6b). This is known as diffraction. Figure 6c now summarizes an important point. In off-axis detection, the length of the detected phasor is zero if the surface is perfectly smooth and no molecules are present on it. The length of the diffracted phasor increases linearly with the amount of molecules. On the other hand, in forward detection, the phasor’s length always stays the same and is equal to the incident phasor. The crux is that photoelectric detection of light can only measure power, hence the phasor’s squared magnitude. This has an important implication on the detection schemes that can be used for diffractometric and interferometric refractometric biosensors (Figure 6).

In general, two main categories of optical signal detection schemes are known [63]: direct detection and indirect (coherent) detection. Direct detection measures the signal’s power and, by this, can only yield the absolute magnitude of the resultant phasor $A_{\mathrm{res}}$:(7)$$I_{\det}={\left|A_{\mathrm{res}}\right|}^{2},$$
where $I_{\det}$ is the detected intensity. Direct detection is simple, but it requires a “strong” signal in order not to be shot noise limited. Coherent detection schemes such as homodyne or heterodyne detection provide a way to translate a phase difference into a change in intensity that can be detected. Thus, interferometric refractometric biosensors will always require an indirect detection scheme, whereas diffractometric biosensors can apply either. Contrary to direct detection, indirect detection schemes require a reference phasor, which is commonly known as the local oscillator $A_{\mathrm{lo}}$. The local oscillator interferes with the signal and produces an interference term that depends on the phase difference between signal and local oscillator:(8)$$I_{\det}={\left|A_{\mathrm{res}}\right|}^{2}+{\left|A_{\mathrm{lo}}\right|}^{2}+2A_{\mathrm{lo}}A_{\mathrm{res}}\cos\left(\Delta \varphi \right).$$

Indirect detection schemes have the advantage that they can provide signal gain if $A_{\mathrm{res}}$ is much smaller than $A_{\mathrm{lo}}$ and by making the detection shot noise limited (meaning that dark current, readout noise, etc. can be suppressed). However, this only considers the detection precision of $I_{\det}$. A reference beam (local oscillator) is never perfectly stable or correlated with the resultant phasor, especially with a longer path difference between reference and signal beams. Thus, any phasor has a relative magnitude and phase uncertainty (Figure 6d) that is proportional to its length. When the signal is already strong enough by itself such that the detection is shot noise limited, it is disadvantageous to have an additional strong reference phasor interfering with a “small” signal phasor. This is because if the reference phasor is not sufficiently stabilized its uncertainty easily masks the small change induced by a biomolecule that binds to the sensor (Figure 6e). This is especially detrimental for amplitude errors because these contribute with the square in Equation (Equation 8) whereas the signal (phase) is only linear. Conversely, direct detection does not require a reference phasor and thus also does not need to stabilize it. In addition, the diffracted beam phasor has a zero length for a perfectly smooth surface because the detection is off axis from the main beam. Even if a small bias is present on the diffraction grating, the diffracted phasor is still small compared with the additional phasor when one molecule binds and can easily be detected. However, the larger the bias, the longer the diffracted phasor relative to the scattered phasor of the additional molecule and the harder it will become to detect it.

Just with this simple picture in mind, one can draw one powerful conclusion (summarized in Figure 6f,g). The fraction of electromagnetic power that hits the detector and carries information about the binding event in interferometric refractometric biosensor (forward and indirect detection) is low and always constant. This is because the incident phasor is large compared to the small (grey) phasor caused by the binding event. In addition, the ratio of incident and resultant phasor is always constant for the same biomolecule. Thus, refractometric biosensors require precise phase stabilization and intensity measurements. This is only possible with sophisticated scientific instrumentation. Conversely, in diffractometric sensors (off-axis and direct detection), the information content is large. This is because the diffracted intensity is much smaller than the incident beam while the scattered phasor has the same length as in forward direction (for TE polarization and the Rayleigh scattering regime) [64]. However, the information content decreases with increasing diffracted intensity (increasing bias mass $\Gamma_{b}$). Therefore, the precision requirement increases until eventually, at a certain bias, it surpasses the one of refractometric biosensors. Thus, diffractometric biosensors will only require less precise measurements as long as their bias is below a certain threshold.

#### 2.5.2. Quantitative Discussion and the Figure of Merit

So far, our discussion was purely qualitative. Everything that was just said can also be rendered quantitative and extended to resonant refractometric sensors by help of the figure of merit of diffractometric and refractometric biosensors. In general, the figure of merit (FOM) stands for how easily a change in sensor output can be detected for a given input [65]. It is easier to detect a difference in sensor output when its absolute change is large (i.e., a high sensitivity). However, a large absolute change does not help if the absolute sensor output is already large/uncertain when the experiment is started. For these reasons, it is not the sensor design with the highest sensitivity that yields the best performance but rather the one with the largest change relative to the uncertainty. The FOM mirrors this in that it is a ratio between the absolute change (sensitivity) normalized to an uncertainty that is specific to the detection mechanism. The larger its value, the lower the measurement precision required to achieve the same mass/refractive index resolution. The effect of relative measurement errors on the sensor output can be described
(9)$$\sigma_{\mathrm{RI}}=\frac{\sigma_{\mathrm{rel}}}{\mathrm{FO}M_{\mathrm{RI}}}.$$

Here, ${\mathrm{FOM}}_{\mathrm{RI}}$ is the figure of merit in refractive index units (RIU), $\sigma_{\mathrm{RI}}$ is the standard deviation in RIU and $\sigma_{\mathrm{rel}}$ is the standard deviation of the relative sensor output. (From Equation (Equation 9), it follows that the figure of merit is equivalent to the sensitivity of the normalized sensor output to refractive index changes provided that the sensor output is in the proper relative unit (relative phase or intensity error, respectively) [16].)

The FOM is best known and most widely used for refractometric sensors that measure phase changes via a resonance condition. A resonance is especially suited to measure the phase because the phase rapidly increases over it by π. Furthermore, a phase reference is no longer required since the phase is defined relative to the location of the resonance in, e.g., wavelength space. (Chapter 9.1 in [64]) Most commonly, the resonance is measured as a shift in coupling angle and via a shift in wavelength or change in intensity [66,67,68,69,70]. For reasonable sensor sizes (interaction lengths) and refractive index changes that occur for mass changes at the detection limit, the phase changes are usually only a fraction of the resonance width [8,66]. The uncertainty is the resonance width. The broader the resonance, the more challenging it becomes to localize its extremum [66]. The resonance width depends on the Q-factor and is characterized by its full width at half maximum (FWHM). The higher the Q-factor, the smaller the FWHM and the higher the FOM [66,69]. However, devices that have a high Q (waveguide grating sensors or whispering gallery biosensors) usually have a low sensitivity, and sensing principles that have a low Q (SPR) have a high sensitivity. Thus, their FOMs are essentially the same [66]. Typically, the FOM is around 50–100/RIU for SPR refractometric biosensors with the very best devices based on narrow Fano resonances reaching 1000/RIU [66,67,68,69,70,71,72,73,74]. (We emphasise that the figure of merits of refractometric biosensors that measure the phase via optical path length differences, e.g., chip-integrated Mach–Zehnder interferometers, are also in the same range [71]. However, the formula stated in that paper is not entirely correct in our view. *P* in the denominator of Equation (Equation 7) should be the total power of the interfering phasors and not the power that is measured at the output. Thus, the FOM will be slightly lower for the chosen operating points.) The FOM of the refractometric readout channel of the sensor in this paper is similar (63.6/RIU (Appendix E)).

While the FOM of refractometric biosensors is well-known and widespread in literature, a FOM for diffractometric biosensors has not yet been formulated. Diffractometric biosensors need to discriminate a change in intensity from the intensity noise of the already diffracted signal before the analyte is applied [27]. For this reason, their FOM is the change in diffracted power (sensitivity) compared to the average of the already diffracted power or stray light (Of course, the average of the already diffracted power is an approximation. The intensity distribution at zero bias is not uniform but rather a fully developed speckle pattern. To properly calculate the uncertainty, one would have to consult a speckle physics book and use the results from first-order statistics of the intensity due to random phasor sums as well as random phasor sums plus a constant reference phasor (Chapter 3.2 in [62]).). Without too much loss of generality, one can attribute a coherent surface mass density to this already diffracted power or stray light—the bias $\Gamma_{b}$. This is allowed as long as the majority of the stray light originates from the sensor surface and therefore has the same coupling constant ($\left|K\right|$) as the molecules of interest [27,31]. Under this assumption, the FOM of a general diffractometric biosensor can be written in an extremely compact form (In a former paper, we denoted this quantity as the sensitivity of focal molography [27]. This is still valid, if the sensor output is normalized to a reference power that has a certain mass attributed to it. The FOM that we introduced in that paper served the purpose of maximizing the signal to background ratio of a waveguide-based molographic arrangement.)
(10)FOMdiff=SΓcohPoutPoutPinPinPout0Pout0PinPin=22KncdndcA2Γb2KncdndcA2Γb2=2Γb.

The most striking difference to the figure of merit of refractometric sensors is that the figure of merit of a diffractometric biosensor is not constant but depends on the output of the sensor—the coherent mass density. This is because the sensor output (intensity) changes quadratically with the sensor input (mass). If the sensor is unbiased, then this residual coherent mass stands for the quality of the background illumination. If the coupling is stronger (larger $\left|K\right|$), the effect of stray light that originates from a different location than the sensor surface will be reduced. This is due to the larger signal per bound molecule. However, in general, the signal to background ratio is only weakly dependent on the sensor configuration. Given that most of the stray light originates from the surface roughness where the molecules of interest are also located [27,31]. Thus, Equation (Equation 10) is valid for any diffractometric biosensing arrangement with a reasonable darkfield illumination and good stray light management [27]. This is an important point because any parasitic stray light interferes with the diffracted signal and hence would compromise the FOM significantly [27].

We have previously shown that the equivalent coherent mass density of the average speckle background of focal molography is only 1.2 pg/mm${}^{2}$[27]. With this value, one obtains a FOM of roughly 2 mm${}^{2}$/pg. If stated in RIU (0.25 pg/mm${}^{2}$≈ 10${}^{-6}$), the FOM${}_{\mathrm{RI}}$ is 500,000/RIU. (There was a unit error for the FOM/sensitivity of focal molography in a previous paper (m${}^{2}$/kg instead of mm${}^{2}$/pg) for which the author apologizes [27]. An erratum has been submitted.) This is three to four orders of magnitude better than the FOM of any refractometric biosensor.

This is because the intensity due to surface roughness and therefore the uncertainty only corresponds to a coherent mass density of 1 pg/mm${}^{2}$ (Figure 7a). On the other hand, refractometric biosensors have a much larger potential uncertainty in the phase measurement (rule of thumb: 1000 pg/mm${}^{2}$ for 2π[8]) (Figure 7b) (The stray light phasor is so “small” because optical surfaces are extremely smooth (0.6 nm standard deviation [27]) and most importantly random with a very short correlation length (10–100 nm). In other words, the height variations of the surface roughness are 1/ξ distributed (where ξ is the spatial frequency) (see the references in [27]). Therefore, the spatial lock-in also helps to reject this noise source [28]. Due to the rapidly oscillating phase of the electromagnetic field on the surface, the phasors due to the roughness have a phase that is uniformly distributed over (−π, π). Thus, they form a random phasor sum, where most of the contributions cancel and the actual resultant phasor length is drawn from the Rayleigh distribution (Chapter 2.2 in [62]). However, one has to be aware that, in a direct measurement, only the magnitude of this phasor can be measured. Therefore, the phase angle between the resultant stray light phasor and the phasor due to the bound molecules is not known. This leads to an uncertainty in the mass quantification that can be computed from the Rician distribution (Chapter 2.3 in [62]). The uncertainty due to the random phasor sum is one of the main disadvantages of direct detection. However, if the application requires it, one can use an indirect technique to determine the phase angle between background and signal phasor.)

The difference in FOM between refractometric and diffractometric biosensors has substantial implications for applications. Let us first consider a resonance-based refractometric sensor with a figure of merit of around 100/RIU (roughly 10${}^{-4}$ mm${}^{2}$/pg): If this refractometric sensor is to exhibit a resolution of $10^{-7}$ RIU or roughly 100 fg/mm${}^{2}$, the location of the resonance needs to be measured with a relative precision compared to its width that is in the order of $10^{-5}$[16]. On the other hand, for the same 100 fg/mm${}^{2}$ mass resolution, a diffractometric biosensor with a bias of 1 pg/mm${}^{2}$ needs a relative precision of the intensity measurement of only 20%. This considerably reduces the requirements on the instrumentation. The simple measurement system described in this paper has a maximum intensity measurement precision of 10${}^{-3}$ over one second (data not shown). Thus, its mass resolution limit of the refractometric output is in the order of 10 pg/mm${}^{2}$, which was achieved for focal distances of 10 mm (Appendix H). For the diffractometric readout, the FOM at a bias of 30 pg/mm${}^{2}$ is 0.066 mm${}^{2}$/pg, which yields a resolution of 150 fg/mm${}^{2}$. If one computes the standard deviation from the baselines in Figure 5, one obtains values of 70–200 fg/mm${}^{2}$. Thus, if the measurement precision is limiting, a diffractometric biosensor has a mass resolution that is, in this example, approximately two orders of magnitude better than the one of the refractometric biosensor.

There is an additional fundamental difference in terms of robustness and ease of readout between refractometric and diffractometric sensors that is worth mentioning. Diffractometric sensors are not only balanced interferometers but also distributed balanced local common path interferometers (Figure 8a,b). Thus, they can be viewed as many pairs of interferometers for which the output is coherently added and thus random phase changes are efficiently rejected. Especially, the effect of long-ranged phase noise is substantially reduced, which brings us back to the spatial lock-in principle. They also do not require splitting of the incident beam into two arms. This is because destructive interference is achieved by the sign change due to oscillation of the phase of the electromagnetic field along the propagation direction. In refractometric sensors, for the same purpose, an accumulated phase shift due to different optical path lengths in two macroscopic interferometer arms is required. This increases the susceptibility to environmental noise. Macroscopic interferometers also require tuning and modulation capabilities to reduce the effect of splitting and source instabilities [7]. Also, the effect of stray light is worse in refractometric devices because scattering in the forward direction is enhanced (Figure 8d) [27].

There are three questions that need to be addressed quantitatively before a conclusive statement can be made on whether diffractometric biosensors offer some unique robustness advantages over refractometric sensors. (1) At what bias will the FOM of a diffractometric biosensor drop below the one of a typical refractometric biosensor? (2) What is the typical dynamic range of a diffractometric biosensor? (3) Are there enough photons at low biases such that in a direct measurement shot noise is not limiting?

To answer the first question, we recall that the required relative precision of the diffractometric measurement increases linearly with the bias (Equation (Equation 10) and Figure 6). For example, a 10 pg/mm${}^{2}$ bias requires 2% relative precision to obtain 100 fg/mm${}^{2}$ resolution. The FOM of a refractometric biosensor is independent of the sensor output. This implies that, above a certain threshold of the bias, the precision of the measurement of a diffractometric biosensor needs to be higher than that of a refractometric device to achieve the same resolution. We derive this threshold for our combined sensor by equating Equation (Equation 28) and Equation (Equation 10) and by solving for $\Gamma_{b}$. For the parameters stated above, this threshold amounts to roughly 7000 pg/mm${}^{2}$. Thus, a mass density of the bias that is higher than a typical monolayer of protein is required to make the diffractometric readout less robust than the refractometric one [56]. For this reason, diffractometric biosensors will have unique advantages when the immobilized molecule is small, such as short oligomers (DNA or peptide). To detect drug binding of small molecules to large immobilized targets [25], either the measurement precision would need to approach that of refractometric devices or a negative bias would need to be introduced in order to balance the diffractometric biosensor. A negative bias could ideally constitute an inactive target molecule or polystyrene, or other nonabsorbing nanoparticles that are immobilized to the grooves.

The second potential problem is dynamic range. Since the diffracted power is quadratically proportional to the amount of bound mass, the diffraction efficiency will quickly attain a maximum value of one. For the quantification formulas to remain valid, one must remain in the quadratic part of the tanh function that describes the diffraction efficiency [46,75]. Therefore, it is beneficial to not increase the diffraction efficiency beyond values of a few percent. From Equation (Equation 2), the dynamic range can be estimated by
(11)$$\Gamma_{\mathrm{coh},\mathrm{DR}}\approx \frac{1}{10\left|K\right|n_{c}\frac{dn}{dc}A},$$
where we have assumed a diffraction efficiency of 4% to obtain a convenient factor of 10 in the denominator. The coupling coefficient *K* might depend on *A* depending on the diffractometric arrangement. For our arrangement and perpendicular outcoupling, the coupling coefficient is $K=4.7\times 10^{14}$/m${}^{3}$. With a circular mologram of diameter 0.4 mm and a refractive index increment of 0.182 mL/g, we obtain a dynamic range of roughly 7000 pg/mm${}^{2}$ (For comparison, a Biacore 8k has a dynamic range of 0.06 RIU (ca. 50,000 pg/mm${}^{2}$). This would require roughly 8 orders of magnitude in the intensity detection capabilities. Six to seven orders of magnitude in exposure time are the state-of-the-art for standard scientific cameras (Grasshopper 3, FLIR with the IMX 250 sensor from Sony). The remainder could be covered by adjusting the laser output or the use of ND (neutral density) filters in the excitation or detection paths. However, dynamic range might become an issue for optical arrangements that have a stronger coupling between incoming and outgoing modes (e.g., coupling between two guided modes) [31]. In conclusion, for most diffractometric arrangements, the dynamic range will be sufficient.

This brings us to the last question: the shot noise limit and whether there are enough photons for direct detection. It can be shown that the shot noise limit on the resolution of a general diffractometric biosensor is given by the following (Appendix F):(12)$$\sigma_{\Gamma_{\mathrm{coh}}}=\frac{1}{4n_{c}\frac{dn}{dc}A\left|K\right|\sqrt{\frac{\eta_{q}\lambda }{hc_{0}}tP_{\mathrm{in}}}},$$
where $\eta_{q}$ is the quantum efficiency of the photodetector, *h* is the Planck constant, $c_{0}$ is the speed of light in vacuum, *t* is the integration time and $P_{\mathrm{in}}$ is the incident power. If we take a beam width of 1 mm, a laser power of 1 mW and an integration time of 1 s and furthermore assume that 90% of the power is lost at incoupling grating, we obtain a resolution of $\sigma_{\Gamma_{\mathrm{coh}}}\approx 1$ fg/mm${}^{2}$ (We took $\eta_{q}=1$ since it does not change the message and even cheap detectors easily achieve $\eta_{q}=0.3$.). In other words, there are always enough photons in diffractometric sensors. In fact, not even the field enhancement provided by a waveguide is strictly necessary (which we previously thought is a requirement) [27]. For the same beam width and mologram sizes, the coupling coefficient between two free space modes is roughly one to two orders of magnitude lower than between a guided and a free space mode (the arrangement described here) [31]. Therefore, with the same laser, the shot noise limit of diffractometric biosensors that couples between two free space modes is at most two orders of magnitude worse—i.e., $\sigma_{\Gamma_{\mathrm{coh}}}\approx 10-100$ fg/mm${}^{2}$, and can be reduced by increasing the laser power. An important lesson should be learned from this. The readout is not photon shot noise limited even for configurations that do not involve field enhancement. Therefore, diffractometric biosensors should be designed for minimal stray light rather than maximum field enhancement. This implies that optimal diffractometric biosensors use the smoothest surfaces and simple and effective optical arrangements. Darkfield illumination is key, and one should employ evanescent waves for the interrogation of diffractometric biosensors (e.g., arrangements that are based on total internal reflection, Bloch surface waves or waveguide modes) [27]. Another important practical factor is that diffractometric biosensors based on outgoing free space modes in combination with array detectors and image registration are essentially self-tuning interferometers. Thus, they are considerably simpler to implement than outgoing guided modes, which will require some kind of tuning mechanism to fulfill the diffraction condition.

## 3. Conclusions

Refractometric biosensors have revolutionized molecular interaction analysis. However, since the introduction of surface plasmon resonance 30 years ago, there was no fundamental technological breakthrough that could address the major problems of refractometric transducers. Sensor equilibration, temperature drifts, buffer change artefacts and nonspecific binding still significantly lower throughput, limit the application scope and complicate the analysis of molecular binding experiments. Most importantly, the stabilization requirements and the cross-sensitivity have impeded label-free (bio)sensors to truly extend their scope beyond the controlled conditions of a laboratory environment. We believe that with the maturation of diffractometric biosensors, not only will this step finally happen but also new applications in the traditional label-free sensing field will emerge. In part I, we provided the theoretical motivation with the concept of the spatial lock-in amplifier [28]. In this paper, we presented the experimental evidence. We demonstrated that diffractometric biosensors are more stable than the best state-of-the-art referenced refractometric biosensors even in the absence of stabilization, sensor equilibration, or sophisticated measurement setups. This is especially true for long measurement times, which might enable unique novel applications, for instance, to measure the off-rates of potent inhibitor drugs as these can have off rates ranging from multiple to thousands of hours [76]. In addition, the combined refractometric and diffractometric biosensors showed that, under the same measurement conditions, diffractometric biosensors can achieve a much higher resolution than refractometric devices. This is all because diffractometric biosensors combine three fundamental principles of a robust biosensor. (1) They form a high-frequency spatial affinity lock-in that performs affinity referencing close to the molecular length scale to effectively separate the binding signal from long-ranged environmental noise [28]. (2) They acquire the signal directly in Fourier space where the binding information of all receptors is concentrated in one point and separated from the noise. (3) They connect the binding event to the magnitude of the electric field rather than the phase or a resonance condition and apply off-axis detection to observe it. This leads to much lower measurement precision requirements for low biases. We believe that, for the first time, these properties will enable miniaturized, multiplexed and simple label-free molecular sensors that are still extremely sensitive and robust. Because of these attributes, spatially locked-in sensors read out by optical diffraction might be the physical principle to emulate the senses of smell and taste of living organisms [77]. It is straightforward to envision a plethora of possible applications of a robust sensor technology for smell and taste ranging from environmental monitoring to process and product surveillance, to security or even military applications, to medical applications in the broad field of diagnostics or smart medical implants and to many more.

## 4. Materials and Methods

### 4.1. Materials

N-Ethyl-N-(3-dimethylaminopropyl)carbodiimide hydrochloride (EDC), sodium dodecyl sulfate (SDS), N-Hydroxysulfosuccinimide sodium salt (NHS) and sNHS-biotin were obtained from Sigma Aldrich. NHS-PEG${}_{12}$ was obtained from Iris Biotech. Carboxyl functionalized polystyrene particles (20 nm) 8.86 × 10${}^{14}$ particles/mL were obtained from Nanocs Inc. (New York, NY, USA) Buffers: MES-T Buffer pH 6: 10 mM MES (2-(N-morpholino)ethanesulfonic acid)), 0.05% Tween20. PB-T pH 7: 100 mM H${}_{2}$NaPO${}_{4}$, 0.05% Tween20. HBS-T pH 7.5: 50 mM HEPES (4-(2-hydroxyethyl)-1-piperazineethanesulfonic acid), 150 mM NaCl, 0.1% Tween20. HBS-T pH 8: 10 mM HEPES, 150 mM NaCl, 0.05% Tween20.

### 4.2. Phase Masks for Mologram Fabrication

Custom phase masks with different numerical aperture (NA) molograms were fabricated by RIE (reactive ion etching) of PECVD (plasma enhanced chemical vapor deposition) deposited SiNx (refractive index: n = 2.14, thickness 320 nm) on 6 mm× 7 mm × 1.1 mm D263 substrates. The chromium dry etch mask were fabricated by e-Beam writing and chromium wet etching. The fabrication process will be described in detail in a later publication. One mask had five different mologram designs (0.33, 0.25, 0.1, 0.05 and 0.02 NA), all with the same diameter of 400 μm. The first row was four 0.33 NA molograms, and the second row was composed of one 0.25, 0.1, 0.05 and 0.02 NA mologram in the direction of mode propagation. The spacing between the center points of the molograms was 500 μm in all directions.

### 4.3. Chip Fabrication for Refractive Index Step Response

The chip was coated with the polymer according to Reference [26] and exposed with 80 mJ/cm${}^{2}$ at 405 nm, as described in [27]. The activation mix for peptide coupling chemistry was prepared by weighting 1.4 mg of EDC and 1.6 mg of NHS into vials and by storing them in the freezer. To activate the mixture, 150 μL MES pH 6 buffer was added (resulting in a 50 mM EDC/NHS solution). We then added 100 μL of the diluted polystyrene particle solution with 5 × 10${}^{13}$ particles/mL to obtain a particle concentration of 2 × 10${}^{13}$ particles/mL in the activation mix. The particles were incubated for 20 min in the dark in the activation mix. The activated particles were mixed with 500 μL of PB-T pH 7 buffer and incubated for 20 min on the chip (final concentration of 7 × 10${}^{12}$ particles/mL). After incubation, we sonicated the chip for 10 min in SDS 2%, rinsed with DI water, isopropylalcohol and DI water again and blow dried. To activate the grooves and surroundings, the chip was flood exposed with a dose of 1200 mJ/cm${}^{2}$ at 390 nm with the setup described in [26] and incubated in 5 mM NHS-PEG${}_{12}$ dissolved in HBS-T pH 8.0 for 15 min. Finally, the chip was rinsed in DI water and blow dried. This protocol produces a chip with a coherent bias mass density $\Gamma_{b}$ of roughly 20 pg/mm${}^{2}$. The polystyrene particles (PS) were adherent to the ridges, whereas the PEG${}_{12}$ was attached to the grooves ([NH-PS|NH-PEG${}_{12}$] molograms).

### 4.4. Chip Fabrication for Streptavidin (SAv) Binding Experiments

The chip was coated with the polymer according to Reference [26] and exposed with 2000 mJ/cm${}^{2}$ at 405 nm, as described in [27]. The activation mix was prepared by weighting 1.4 mg of EDC and 1.6 mg of NHS into vials and by storing them in the freezer. To activate the mixture, 150 μL MES pH 6 buffer was added (resulting in a 50 mM EDC/NHS solution). We then added 100 μL of the diluted polystyrene particle solution with 1.5 × 10${}^{13}$ particles/mL to obtain a particle concentration of 6 × 10${}^{12}$ particles/mL in the activation mix. The particles were incubated for 20 min in the dark in the activation mix. The activated particles were mixed with 500 μL of PB-T pH 7 buffer and incubated for 20 min on the chip (final concentration 2 × 10${}^{12}$ particles/mL). After incubation, we sonicated the chip for 10 min in SDS 2%; rinsed it with DI water, isopropylalcohol and DI water again; and blow dried it. Subsequently, the chip was incubated in 500 μL sNHS-biotin in HBS-T pH 8.0 for 15 min and, after that, rinsed with DI water and blow dried. To activate the grooves and surroundings, the chip was flood exposed with a dose of 1200 mJ/cm${}^{2}$ at 390 nm with the setup described in [26] and incubated in 5 mM NHS-PEG${}_{12}$ dissolved in HBS-T pH 8.0 for 15 min. Finally, the chip rinsed in DI water and blow dried. This protocol produces a chip with a coherent bias mass density $\Gamma_{b}$ of roughly 5 pg/mm${}^{2}$ and has biotins on the ridges that can bind SAv. The polystyrene particles (PS) and the biotin were adherent to the ridges, whereas the PEG${}_{12}$ was attached to the grooves ([NH-biotin + NH-PS|NH-PEG${}_{12}$] molograms).

### 4.5. Refractometric and Diffractometric Signal Quantification

The basic image processing algorithm for focal molography was described in [27] Figure 27. We slightly adapted the algorithm in that we used the subpixel image registration implemented in CalmAn [78] based on the image processing library OpenCV. Furthermore, we used the total outcoupled power measured by a photodiode and corrected by the exponential decay in the waveguide to compute $P_{\mathrm{in}}$ (compared to the average background intensity in [27]). The shift computed by the image registration algorithm was used to compute the refractometric signal.

### 4.6. Measurement of Refractive Index Step Response

*Combined refractometric diffractometric sensor*: Six refractive index standard solutions were obtained by mixing Glycerol (Sigma Aldrich) and MilliQ (DI) water in a ratio predicted from the Gladstone-Dale model [79]. The refractive index of the six solutions increased approximately in steps of 0.005 RIU, and the precise refractive index of the solutions was measured by a benchtop refractometer (J457 Refractometer, Rudolph Research Analytical). The experiment was performed with a 0.4 NA objective and a 0.33 NA mologram.

Biacore 8k: A CM5 (carboxymethylated dextran) chip was used, and the same six glycerol/water refractive index solutions were injected at a flow rate of 30 μL/min into a signal and a reference channel. MilliQ water was used as a running buffer. For the conversion from RU to pg/mm${}^{2}$, the assumption 1 RU ≈ 1 pg/mm${}^{2}$ was made.

### 4.7. Resolution Measurement

The experiment on the Biacore 8k+ was conducted with the following parameters. The 8k+ instrument was cleaned with the usual maintenance protocols: desorb and sanitize, desorb and superclean. PBS-T pH 7.4 containing 11 mM PO${}_{4}$, 137 mM NaCl, 2.7 mM KCl and 0.05% Tween20 was used as the running buffer. A new CM5 chip and a 2.5-month in-place IFC (Integrated Microfluidic Cartridge) were used for the measurements. The running buffer was injected for one minute as the analyte and the dissociation phase was measured for 200 min at 25 ${}^{\circ }$C with a data collection rate of 1 Hz. The flow rate was varied to 10, 50 and 100 μL/min. The evaluation of the data was performed using Insight SW 3.0.11.15423, and the despiked data were exported. The experiment was aligned to the report point “stability early”. Each experiment was performed in triplicates (three cycles). For evaluation, the best and the worst channels of the eight channels of each experiment were used. For focal molography, the experiment was carried out on a chip fabricated as described in [26,27] with the flow cell described in [27]. The coherent surface mass density was calculated from the outcoupled power. The power at the location of the mologram was 0.045 W/m. The sampling frequency was 1.6 Hz. The diameter of the mologram was 0.4 mm, and the numerical aperture was 0.1. The microscopy objective had a numerical aperture of 0.1 as well. The buffer was filtered and degassed wtih PBS-T (ThermoFisher, 0.05% Tween20 added), and the flow rate was 5 μL/min. The molograms were weak SAv molograms (2–2.5 pg/mm${}^{2}$).

### 4.8. Intensity Conversion

We have never mentioned the intensity conversion from a binary image in any of our previous papers and shall quickly state this here. Here, we also used a different camera model compared to [27] in the moloreader (model: GS3-U3-50S5M-C, FLIR Systems, Inc. with the IMX250 image sensor from Sony). The intensity was obtained from the photon flux ϕ: $I=\phi \frac{hc}{\lambda }$, with *h* as the Planck constant and *c* as the speed of light in a vaccum. The photon flux was obtained from $\phi =\frac{M^{2}n_{p}}{A_{p}t}$, with M as the magnification of the optical system, $A_{p}$ as the area of one pixel, *t* as the exposure time and $n_{p}$ as the number of photons. The number of photons was obtained from the binary image $n_{p}=\frac{\mathrm{bin}\_\mathrm{image}}{p_{\mathrm{Gain}}\eta }$ with η as the quantum efficiency and $p_{\mathrm{Gain}}=\frac{\max\_\mathrm{bin}\_\mathrm{value}}{FWC}$, which is the ratio of the maximum binary value (here 4095) and the full well capacity (FWC) (in number of electrons, for this camera, 10,361 e${}^{-}$). In general, one has to be careful that the camera gain is set accordingly for this conversion to be valid. As a side remark, the output of the implemented image processing algorithm is the Airy disk convolved intensity $I_{\mathrm{sig}}$, which is related to the total diffracted power by $I_{\mathrm{sig}}=1.69\frac{P_{\mathrm{out}}}{A_{\mathrm{Airy}}}=1.45\frac{P_{\mathrm{out}}NA^{2}}{\lambda^{2}}$ [27].

## Figures and Tables

**Figure 1 sensors-21-00009-f001:**
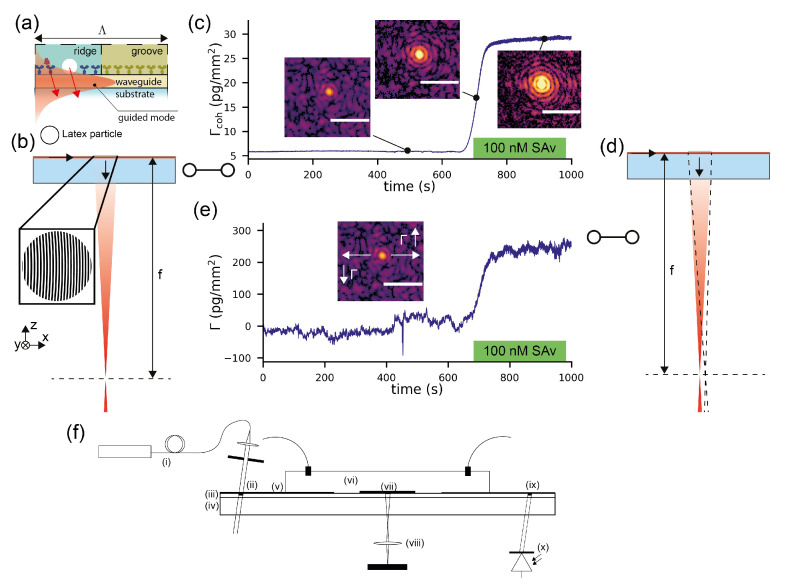
Concept of a waveguide-based combined refractometric and diffractometric biosensor for demonstration of the robustness of spatially locked-in diffractometric biosensors: (**a**) the sensor constitutes a diffraction grating on top of a high refractive index Ta${}_{2}$O${}_{5}$ slabwaveguide. The diffraction grating consists of roughly 1000 ridges and grooves with grating period Λ. One grating period is enlarged. A small bias (mass excess) on the ridges is introduced by chemically linked polystyrene particles (Latex) that yield a detectable diffraction signal. An antibody or similar capture molecule (in this case, just biotin) is immobilized on the ridges that bind the target molecule. The light diffracted by the target molecule interferes constructively with the light diffracted by the polystyrene particles. The grooves were functionalized by a molecule that is chemically similar to the capture molecule on the ridges but does not bind the target molecule (in this case, a short polyethylene glycol) [26,46]. (**b**) The diffraction grating has the form of a diffractive lens that focuses the diffracted light from the guided mode at a focal distance *f*. (**c**) Experimental diffractometric signal: as molecules bind (Streptavidin—SAv), the diffraction efficiency of the grating increases, which leads to a quadratic increase of the diffracted power with bound mass. From this, a binding curve can be generated. The diffractometric biosensor measures the refractive index difference, i.e., the coherent mass density $\Gamma_{\mathrm{coh}}$, between ridges and grooves. [27,31,50] (an image of the focal plane is shown in the insets; the scale bar is 30 μm). (**d**) The binding of molecules, irrespective of whether they bind on the ridge or the groove, increases the average refractive index of the cover medium close to the waveguide surface. This causes a change in the effective refractive index of the guided mode and hence in the outcoupling angle of the diffracted light, which manifests as a shift of the focal spot in the focal plane in the mode propagation direction. A binding event to the grooves cannot be discriminated from one to the ridges. (**e**) Experimental refractometric signal: from the shift of the molographic focal spot, a binding curve can be generated. The refractometric sensor measures the integrated refractive index change in the entire sensing volume, i.e., the total surface mass density Γ. (**f**) Schematic of the readout system: (i) a fiber coupled laser is expanded and hits a coupling grating (ii), sitting on the surface of the (iii) waveguide attached to a transparent substrate (iv). The grating is protected by an SiO${}_{2}$ cover (v). (vi) Fluidics with connectors to execute real-time measurements. (vii) An array of molograms and (viii) the optical system observe them. An outcoupling grating (ix) in combination with a photodiode (x) quantifies the incident power.

**Figure 2 sensors-21-00009-f002:**
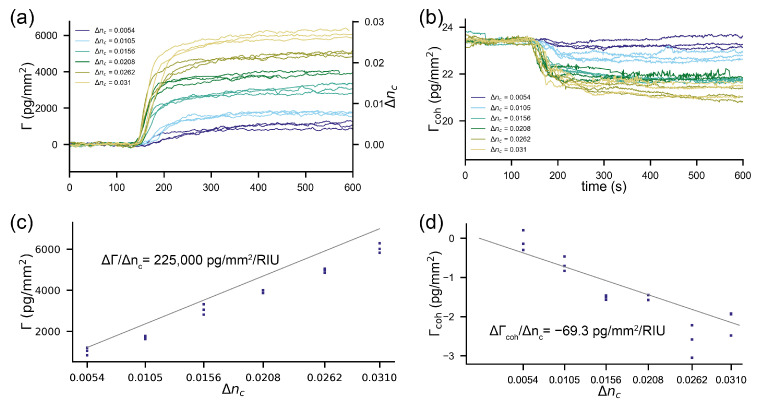
Cross-sensitivity to refractive index discontinuities of the two readout channels of the combined sensor: the curves are not step responses due to diffusive mixing within the suboptimal fluidics of the device. (**a**) Refractometric readout channel: The signal is displayed in both RIU (refractive index units) as well as pg/mm${}^{2}$ (surface mass density Γ). (**b**) Diffractometric readout channel. (**c**) The experimentally measured change in surface mass density Γ as a function of the applied refractive index change $\Delta n_{c}$: the grey line is a theoretical prediction from Equation (Equation 4). (**d**) The experimentally measured change in coherent mass density $\Gamma_{\mathrm{coh}}$ as a function of the applied refractive index change $\Delta n_{c}$ for the diffractometric sensor output: the grey line is a theoretical prediction from Equation (Equation 5).

**Figure 3 sensors-21-00009-f003:**
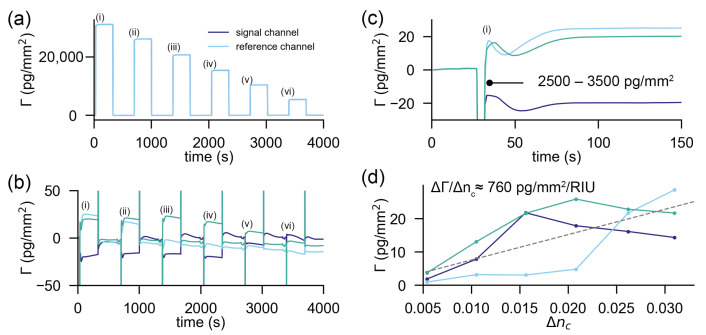
Response of a state-of-the-art Biacore 8k instrument to refractive index discontinuities: (**a**) signal and reference channel responses to the subsequent injections of six solutions with decreasing refractive index (i) 1.3640, (ii) 1.3592, (iii) 1.3528, (iv) 1.3486, (v) 1.3435 and (vi) 1.3384) into MilliQ (*n* = 1.333) as a running buffer (the signal and reference channels are overlapping in this plot). (**b**) Response of the reference subtracted signal trace for three experiments. (**c**) Zoom into the first refractive index jump: the time delay between the two channels causes a considerable discontinuity in the sensor response followed by a transient of roughly 50 s. From then on, the discrepancy is solely due to different penetration depths in signal and reference channels. (**d**) Refractive index errors at the end of the transient response of each applied refractive index discontinuity as a function of the change in cover index. The dashed line is the linear fit to the data of the three experiments, and its slope equals the sensitivity to refractive index jumps of the referenced refractometric sensor. The discontinuities at the beginning and end of the refractive index jumps due to time delays in the reference and signal channel were excluded in the analysis.

**Figure 4 sensors-21-00009-f004:**
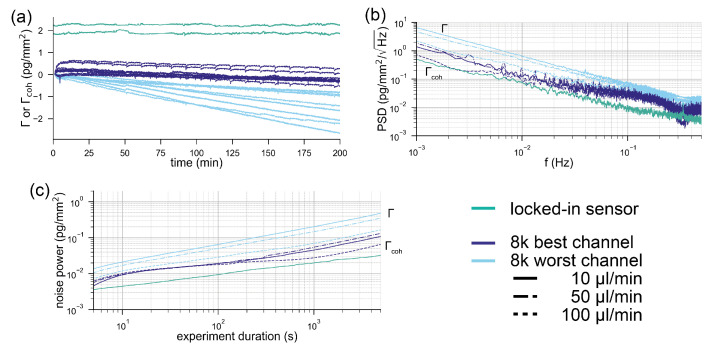
Comparison of the noise performance of a state-of-the-art referenced refractometric biosensor with a spatially locked-in diffractometric sensor: (**a**) typical baselines of the locked-in diffractometric readout channel (green) extracted from two molographic spots and the response of the best (dark blue) and the worst channel (light blue) from the eight channels of a Biacore 8k+ instrument. The displayed response is the difference between flow cell 1 and flow cell 2 (referenced binding trace). In a Biacore 8k+, each of the eight channels has two flow cells operated in parallel and the referencing between two flow cells is considered crucial in high-performance measurements. In total, nine baselines are shown (three different flow rates and triplicates for each experiment). The unit for the molography traces is the coherent surface mass density, whereas for the Biacore system, it is surface mass density. (**b**) Power spectral density of the binding traces shown in (**a**) the average power spectral density of the triplicates for each experiment was computed and displayed. For the diffractometric readout channel, the power spectral density was computed from one trace, since it was almost identical for both traces. (**c**) Integrated power spectral densities of the surface mass density/coherent surface mass density (noise power picked up by the system) for different experiment durations.

**Figure 5 sensors-21-00009-f005:**
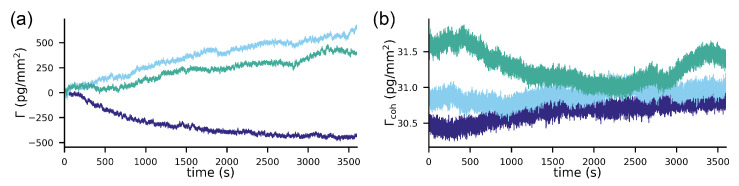
Comparison of the resolution of an unreferenced refractometric and a diffractometric biosensor subject to the very same experimental conditions: (**a**) refractometric output for three different experiments in PBS-Tbuffer at a flow rate of 5 μL/min with 0.1 numerical aperture (NA) molograms (2-mm focal distance). (**b**) Simultaneous diffractometric output for the same three experiments: the noise of the diffractometric baselines is roughly three orders of magnitude smaller than the refractometric one.

**Figure 6 sensors-21-00009-f006:**
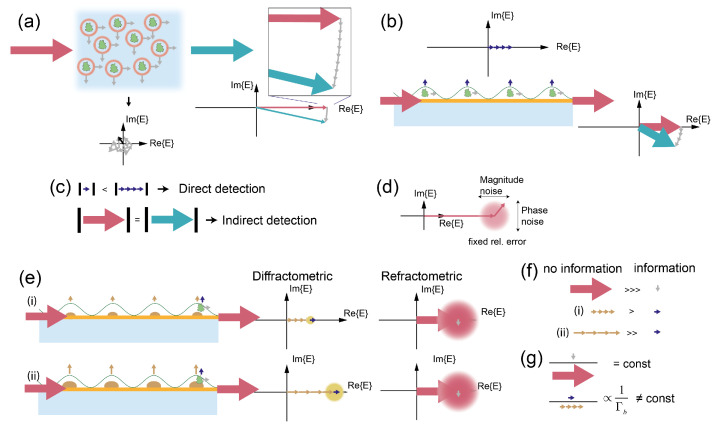
Fundamental differences in interferometric refractometric and diffractometric detection of biomolecular binding and their implications on readout precision: (**a**) a dilute ensemble of biomolecules is illuminated by coherent light. The molecules scatter light in all directions. In the forward direction, the interference between the incident and scattered fields results in phase retardation of the incident phasor (refraction). In off-axis directions, the scattered phasor is a random phasor sum, and most importantly, the main beam does not interfere with the scattered phasors. (**b**) Molecules on a substrate are arranged such that their scattered fields interfere constructively in the direction of the surface normal (diffraction). The length of the individual phasors add up and fully contribute to the resultant phasor. The effect in the forward direction is still only a rotation of the incident phasor with no magnitude change. (**c**) For off-axis detection (diffraction), the length of the resultant depends on the amount of molecules. This allows for direct detection methods. For forward detection (refraction), the length of the resultant phasor is not changed and hence can only be detected with indirect methods (homodyne and heterodyne detection) that supply a phase reference. (**d**) Every phasor that can interfere at the detector can only be stabilized/measured to/with a certain precision. This leads to an uncertainty in magnitude and phase of the phasor. (**e**) (i) A surface-based biosensor with a low bias (yellow) and the effect of one additional molecule (green) that binds to it: The diffractometric as well as the refractometric sensor responses have an uncertainty that is proportional to the length of the phasor that hits the detector before the measurement. Since the refractometric response includes the incident (strong) beam, the small scattering phasor due to the molecule cannot be detected. This is possible in the diffractometric response because the already diffracted power is comparable to the magnitude of the additional phasor. (ii) The same sensor with a larger bias (higher diffraction efficiency): while the relative uncertainty remains constant, the absolute value increases. If the relative measurement precision is the same as before, the diffractometric sensor can no longer detect binding of the molecule. (**f**) The information content of the electromagnetic fields is another way to explain the higher relative measurement precision requirements of refractometric sensors. In refractometric sensors, the large incident phasor does not contain any information of the binding event. Only the small grey phasor contains binding information. In contrast, the information content of diffractometric biosensors is much higher as long as the diffracted phasor before the molecule binds is not too large. (**g**) The information content about the binding event for given refractometric sensors is always constant. For a diffractometric biosensor, it is inversely proportional to the coherent mass that is already on the sensor.

**Figure 7 sensors-21-00009-f007:**
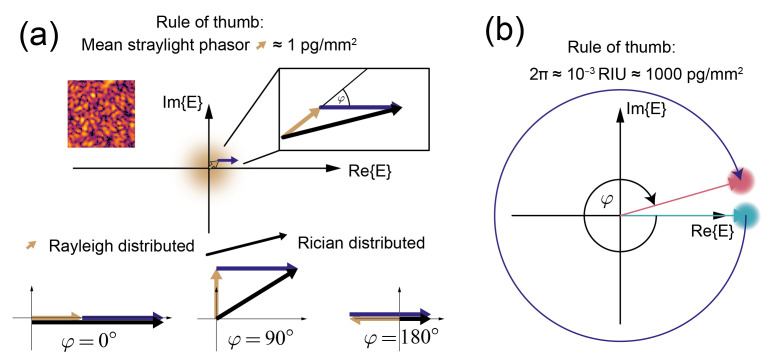
Rules of thumb for the lengths of typical stray light phasors of diffractometric biosensors as well as the refractometric phasors phase change per bound mass: (**a**) the stray light phasor length for optically smooth surfaces typically corresponds to 1 pg/mm${}^{2}$ and is Rayleigh distributed. By interference in the far field, all roughness phasors form one random phasor sum with resultant phasors that form a fully developed speckle pattern with a negative exponential distribution of intensity. Thus, even a 10% magnitude error of the phasor results in only a 50 fg/mm${}^{2}$ mass density error (Equation (Equation 10) and the mass density version of Equation (Equation 9)). In a direct measurement, the phase angle between stray light (yellow) and the phasor of the bound molecules (violet) is not known. The length of the resultant (black) phasor that is measured therefore follows a Rician distribution, and its length depends on the mutual orientation of stray light and signal phasors. The figure shows the resultant phasor’s length for three different phase angles. This uncertainty can be overcome by measuring the phase angle φ between the stray light phasor and the signal phasor by a phase reference (indirect technique). (**b**) In good refractometric biosensors, roughly 1000 pg/mm${}^{2}$ corresponds to a phase change of 2π. Thus, a measurement error of 10% will give rise to a 100 pg/mm${}^{2}$ error in terms of mass density.

**Figure 8 sensors-21-00009-f008:**
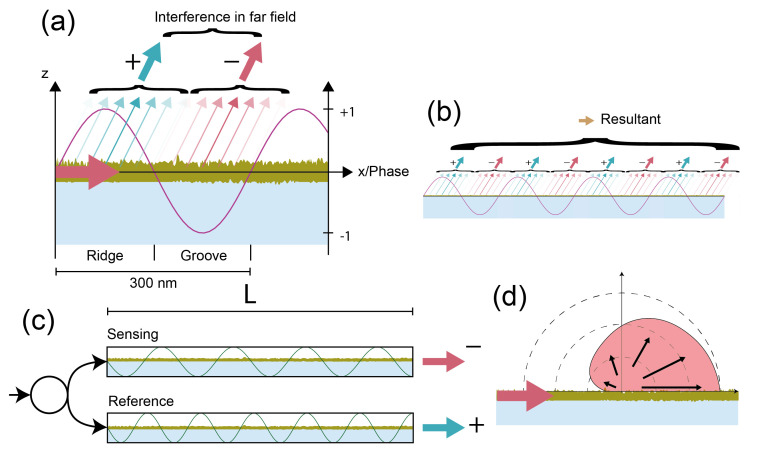
Diffractometric biosensors as off-angle local common path vs. refractometric interferometers: (**a**) diffractometric biosensors are a nearly perfectly balanced off-angle local common path interferometer. Oscillation of the phase of the electromagnetic field defines regions of constructive and destructive interference for a given diffraction condition. Therefore, the contribution to the resultant phasor varies sinusoidally over one grating period, whereas the edges of the ridges and grooves hardly contribute. (**b**) In the farfield, the contribution of all reference and signal regions is added up interferometrically to one resultant phasor. (**c**) In interferometric refractometric biosensors, the phase change is linked to different optical path lengths rather than defined by oscillation of the electromagnetic field. For this, the beam needs to be splitted, which introduces noise. (**d**) Scattering on the surface of a waveguide is anisotropic since the roughness power spectral density is typically Lorenzian [27]. This higher noise at lower spatial frequencies leads to predominant forward scattering and hence to worse stray light interferences in interferometric refractometric than diffractometric biosensors.

## Data Availability

The data presented in this study are available on request from the corresponding author. The data are not publicly available since all relevant data can be extracted from the figures.

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
