# Peer review of "Ultra Stable Molecular Sensors by Submicron Referencing and Why They Should Be Interrogated by Optical Diffraction—Part II. Experimental Demonstration"

_sensors, 2020, doi:10.3390/s21010009_

Round 1

Reviewer 1 Report

This manuscript presents theoretical explanations and experimental validations of the diffractometric biosensing methodology. Compared with well-known refractometric biosensing method, the developed diffractometric biosensing method offers better stability and resolution. Based on the proposed theoretical explanations, the authors address three factors: (1) spatial lock-in mechanism; (2) direct measurement via spatial convolution; (3) off-axis detection. In general, this is a good manuscript for publication. However, it could be improved for general readers with following comments.

  1. Sometimes, there are too many clauses within a sentence. Just have to read twice to get the idea. It is suggested that the author could have concise words for statements.
  2. Fig. 6 is important in this manuscript, however, it is not straight forward. There are too many arrows, e.g. sizes, colors, and numbers. It is suggested the author should revise the figure with a better representation.
  3. To help reader to have a clear picture, it is suggest the author could consider a schematic of the experimental set up. It would help to readers to catch the idea.

Author Response

See attached pdf.

Reviewer 2 Report

The article presents a demonstration of the superior performance of a spatially locked-in diffractometric sensor over the conventional refractometric SPR system.

The results are very interesting and the work is well presented.

I recommend the publication of the work as it is

Author Response

See attached pdf.

Reviewer 3 Report

This manuscript reports a thorough investigation of the use of diffraction as the basis for optical transduction of binding events on appropriately modified sensor surfaces. The authors combine both empirical results, including from comparison with a sophisticated surface plasmon resonance instrument, with theoretical characterisation of the critical parameters of a diffraction-based transduction system. Extensive comparison is made to well-established refraction-based approaches and the authors make a reasonable argument for how diffraction offers some fundamental advantages over more traditional approaches. It appears from the analysis presented that diffraction-based transduction could, given appropriate care with surface design and modification and use of appropriate biorecognition elements, become an increasingly useful approach in future. The manuscript is generally well-written and very amply detailed. The figures provided support the analysis in the text. There are however some points that I feel the authors could address and these are detailed below.

  1. Page 1 lines 13-15 “. . . we believe that . . . smart devices.” Whilst it is reasonable to extrapolate a bit into the future concerning the potential for new sensor approaches, the authors should bear in mind that the transduction method is not the only critical factor determining sensor performance in the field. Surface chemistry, the type of recognition element used and the composition of the samples used, including their background matrix, are all critical determinants of in-field sensor performance.
  2. Page 7 lines 204-205 “One can see . . . by 2-3 pg/mm2.” It would be useful to include the slopes of the plots in Figure 2c and d to allow for a clearer comparison of this feature.
  3. Figure 4 legend. The best channel in Figure 4a is in fact the dark blue one and the worst is the light blue. You have this labelling correct in the figure but incorrect in the figure legend.
  4. Figure 4 legend. “. . . surface mass density / coherent surface mass density . . .” Using a “/” here may give the erroneous impression that you suggesting a ratio of the two densities here. To avoid confusion you should replace the “/” with “or”.
  5. Page 11 lines 298-299 “The refractometric sensing . . . rms noise).” How precise can you be about the relative rms noise values? Are they only quoted to the nearest order of magnitude or can you define this more exactly?
  6. Page 22 line 558 “. . . different NA molograms . . .” For clarity, define this abbreviation at this point in the manuscript.
  7. Chip Fabrication in the Materials and Methods section. For ease of reading, the authors should make it clear in the manuscript how the surface of the chip was differentially modified so the polystyrene particles adhered on the ridges but the PEG12 adhered to the grooves.

Author Response

See attached pdf.
